# Efficient Removal of Non-Structural Protein Using Chloroform for Foot-and-Mouth Disease Vaccine Production

**DOI:** 10.3390/vaccines8030483

**Published:** 2020-08-27

**Authors:** Sun Young Park, Jung-Min Lee, Ah-Young Kim, Sang Hyun Park, Sim-In Lee, Hyejin Kim, Jae-Seok Kim, Jong-Hyeon Park, Young-Joon Ko, Choi-Kyu Park

**Affiliations:** 1Animal and Plant Quarantine Agency, Gimcheon-si 39660, Korea; sun3730@korea.kr (S.Y.P.); wjdalslee@korea.kr (J.-M.L.); mochsha@korea.kr (A.-Y.K.); shpark0205@korea.kr (S.H.P.); lunark2@korea.kr (S.-I.L.); hyejin86j@korea.kr (H.K.); kimjs0728@korea.kr (J.-S.K.); parkjhvet@korea.kr (J.-H.P.); 2College of Veterinary Medicine, Animal Disease Intervention Center, Kyungpook National University, Daegu 41566, Korea

**Keywords:** foot-and-mouth disease virus, inactivated vaccine, non-structural protein, purity, chloroform

## Abstract

To differentiate foot-and-mouth disease (FMD)-infected animals from vaccinated livestock, non-structural proteins (NSPs) must be removed during the FMD vaccine manufacturing process. Currently, NSPs cannot be selectively removed from FMD virus (FMDV) culture supernatant. Therefore, polyethylene glycol (PEG) is utilized to partially separate FMDV from NSPs. However, some NSPs remain in the FMD vaccine, which after repeated immunization, may elicit NSP antibodies in some livestock. To address this drawback, chloroform at a concentration of more than 2% (*v/v*) was found to remove NSP efficiently without damaging the FMDV particles. Contrary to the PEG-treated vaccine that showed positive NSP antibody responses after the third immunization in goats, the chloroform-treated vaccine did not induce NSP antibodies. In addition to this enhanced vaccine purity, this new method using chloroform could maximize antigen recovery and the vaccine production time could be shortened by two days due to omission of the PEG processing phase. To our knowledge, this is the first report to remove NSPs from FMDV culture supernatant by chemical addition. This novel method could revolutionize the conventional processes of FMD vaccine production.

## 1. Introduction

Foot-and-mouth disease (FMD) is a highly contagious viral disease of cloven-hoofed animals. It is the most serious livestock disease due to the economic consequences of its outbreak. The causative agent, FMD virus (FMDV), belongs to the *Aphthovirus* genus of the family *Picornaviridae*. The virus exists in the form of seven different serotypes: O, A, Asia 1, C, and South African Territories (SAT) 1, SAT 2, and SAT 3 [1]. A single-stranded positive RNA virus genome encodes four structural proteins (SPs; VP1, VP2, VP3, and VP4) and several non-structural proteins (NSPs; L, 2A, 2B, 2C, 3A, 3B, 3C, and 3D) [2]. The FMDV rapidly replicates and spreads within the infected animal, among in-contact susceptible animals, and through aerosol [3].

South Korea has experienced 11 FMD outbreaks since 2000. In particular, massive FMD outbreaks in 2010–2011 prompted the government to adopt vaccination policies across the country. Since then, FMD-susceptible livestock have been vaccinated with vaccines imported from the UK, Argentina, and Russia. The government is trying to develop an FMD vaccine using local strains isolated during outbreaks.

The currently-used FMD vaccine is an inactivated virus formulation prepared in a suspension of baby hamster kidney-21 (BHK-21) cells [4]. The use of inactivated virus ensures safety against FMD virus transmission and is suitable for DIVA (differentiating infected from vaccinated animals) strategy application. Because inactivated FMDV cannot replicate due to the loss of infectivity, non-structural proteins (NSPs) are not produced in animals vaccinated with FMD vaccine. Accordingly, the antibodies against SPs are induced in both vaccinated and naturally infected animals, whereas the antibodies against NSPs are only produced in animals naturally infected with FMDV. Therefore, one of the important elements in FMD vaccine antigen production is the elimination of viral NSPs.

For this reason, unlike other animal vaccines, FMD vaccine requires a purification step during the manufacturing process. As a method for concentration and purification of inactivated crude FMDV culture supernatant, ultrafiltration, chromatography, and PEG precipitation methods are described in the World Organization for Animal Health (OIE) terrestrial manual [5].

In general, PEG is the most widely used for FMD vaccine production [6]. PEG is a nonionic polymer causing dehydration in the aqueous layer surrounding the protein, thereby increasing the binding capacity of the protein to another protein and consequently causing its precipitation. However, vaccines made with this PEG method inevitably contain trace amounts of NSPs [7,8]. In such cases, residual NSP could induce NSP antibodies in some livestock after repeated vaccination [9].

Therefore, the purpose of this study was to develop a new method to more efficiently and simply remove NSPs from the FMDV culture supernatant by chemical treatment.

## 2. Materials and Methods

### 2.1. Cells and Viruses

Baby hamster kidney-21 (BHK-21) suspension cells were developed from the original adherent cell line, BHK-21 (C-13) (ATCC, Manassas, VA, USA) by the Animal and Plant Quarantine Agency and the Korea Research Institute of Bioscience & Biotechnology for use in suspension culture with serum-free media [10]. The cells reached at least 1.5 × 10^6^ cells/mL from a seeding density of 3 × 10^5^ cells/mL within 72 h. The type O FMDV strains (O/Boeun/SKR/2017 and O/Andong/SKR/2010), isolated in Korea, were used to inoculate BHK-21 suspension cells (3 × 10^6^ cells/mL) at a multiplicity of infection (MOI) of 0.001 on a shaking platform in an incubator held at 37 °C. Viruses were harvested at 16 h post-infection and clarified by centrifugation at approximately 3000× *g* for 20 min at 4 °C to remove cell debris. Binary ethylenimine (BEI, Sigma-Aldrich, St. Louis, MO, USA) was added at 3 mM to the virus culture supernatant to inactivate FMDV, followed by incubation at 100 rpm for 28 h at 26 °C. Subsequently, the BEI was neutralized by adding 2% sodium thiosulfate (Daejung Chemicals & Metals, Siheung, Korea).

### 2.2. Chloroform Treatment

The inactivated type O FMDV culture supernatant was treated with various concentrations (0–10%) of chloroform (Merck KGaA, Darmstadt, Germany) and mixed by inverting for 5 min at room temperature. The mixture was centrifuged at approximately 3000× *g* for 15 min, and then the aqueous layer on top of the organic solvent was harvested. As a control, trichloroethylene (TCE), structurally similar to chloroform, was employed and treated similarly.

### 2.3. Western Blot Analysis

FMDV samples were mixed with 4× Lithium dodecyl sulfate (LDS) sample buffer (Invitrogen, Carlsbad, CA, USA). The pretreated samples were heated at 70 °C for 10 min. Samples were run on 4–12% gradient bis–tris gels (Invitrogen) transferred to a nitrocellulose membrane using the iBlot2 gel-transfer device (Invitrogen). The membranes were blocked in phosphate-buffered saline with Tween-20 (PBST, 10 mM sodium phosphate, 132 mM NaCl, 2.7 mM KCl, and 0.05% Tween-20, pH 7.4). The membranes were washed and incubated with anti-FMDV type O VP1 (76.5E) and anti-FMDV 3B (4G24) monoclonal antibodies, produced in-house, diluted 1/2000 in PBST, followed by incubation with goat anti-mouse horseradish peroxidase-conjugated secondary antibodies (Invitrogen) diluted 1/4000 in PBST. Proteins were visualized with Pierce ECL Substrate (Invitrogen) using an Azure C600 imaging system and the cSeries Capture Software (Azure Biosystem, Dublin, CA, USA).

### 2.4. Virus Titration

Viral titers were determined by endpoint titration. Serum-free Dulbecco’s modified Eagle medium (DMEM, Invitrogen) was added to each well of a 96-well plate. Then, 10-fold serially diluted infective FMDV and adherent BHK-21 cells were added to the plates in the same volume. The plates were incubated in an atmosphere of 5% CO_2_ at 37 °C for 3 days. The cytopathic effect was observed under a microscope, and the titer was determined according to the Spearman–Kärber method [11]. The viral titer was expressed as log 10 TCID_50_ per milliliter.

### 2.5. Preparation of Vaccines

FMD vaccines were prepared with (i) PEG-precipitated and (ii) chloroform-treated FMD vaccine antigens. The inactivated viral culture supernatant was obtained by the method described in Section 2.1, and the following procedures were performed. For the first group, the supernatant was treated with 7.5% (*w/v*) PEG 6000 (Sigma-Aldrich), stirred overnight at 4 °C, and centrifuged at 10,000× *g* for 30 min. After the supernatant was removed, the pellet was resuspended in tris–KCl buffer (pH 7.6) and adjusted to a concentration of 15 μg/mL. For the second group, the supernatant was mixed with 10% (*v/v*) chloroform by inverting for 5 min, and centrifuged at approximately 3000× *g* for 15 min. The aqueous layer on the top of the organic solvent layer was harvested and concentrated to a final concentration of 15 μg/mL by an ultrafiltration device fitted with a polyethersulfone membrane with a MWCO (molecular weight cutoff) of 300 kDa (Millipore, Billerica, MA, USA). Saponin (Sigma-Aldrich) and aluminum hydroxide gel (General Chemical, Parsippany, NJ, USA) were added to these two types of antigens, and then the ISA 206 VG adjuvant (Seppic, Paris, France) pre-warmed at 30 °C was added in a ratio of 1:1. After the mixtures were blocked from light and incubated at 20 °C for 1 h in a water bath, they were stored at 4 °C until use.

### 2.6. Immunization of Animals

Korean black goats at the age of 8 months, which were not previously FMD-vaccinated, were tested by enzyme-linked immunosorbent assay (ELISA) to confirm negativity for SP and NSP antibodies. A total of 20 goats were injected intramuscularly with vaccines containing PEG (*n* = 10) and chloroform (*n* = 10) treated antigens. All goats were vaccinated 3 times every 4 weeks at a dose of 2 mL and bled at 0, 4, 8, and 12 weeks after the initial vaccination to obtain serum samples. The animal experiments in this study were approved by the Institutional Animal Care and Use Committee (IACUC) and carried out in accordance with the National Institutes of Health Guide for the care and use of laboratory animals (IACUC approval no. 2020-542).

### 2.7. ELISA

PrioCHECK FMDV NSP and type O SP antibody detection kits (Prionics, Lelystad, The Netherlands) were used to measure antibody levels elicited by the vaccines. ELISA was performed according to the manufacturer’s instructions. Test sera were added to antigen-coated plates and incubated for 1 h at room temperature. After the plates were washed, horseradish peroxidase-conjugated secondary monoclonal antibodies were added to the plates, followed by incubation for 1 h. Finally, the plates were washed and 3,3′,5,5′-tetramethylbenzidine substrate solution was added. The colorimetric reaction was stopped with sulfuric acid solution after 15 min, and the optical density (OD) was measured at 450 nm. The ELISA results were expressed as percentage inhibition (PI) values. Samples with a PI value of ≥50% were considered positive, and those <50% were considered negative.

### 2.8. Virus Neutralization Test

The virus neutralization test (VN) for the FMDV antibody was performed using the methods described in the OIE terrestrial manual [5]. The sera were inactivated at 56 °C for 30 min before testing. The 50 μL of two-fold serially-diluted sera starting from 1/4 dilution was mixed with 50 μL of virus (FMDV O/Boeun/SKR/2017) containing 100 TCID_50_ (50% tissue culture infective dose). After incubation at 37 °C for 1 h, 50 μL of fetal porcine kidney (LFBK, supplied by Plum Island Animal Disease Center (Orient, NY, USA)) cells (10^6^ cells/mL) were added to each well. Plates were sealed and incubated in an atmosphere of 5% CO_2_ at 37 °C for 2–3 days. The cytopathic effect was observed, and the VN titer was determined according to the Spearman–Kärber method [12]. The VN antibody titer was converted to log 10 scale for graphical purposes. A titer of 1:45 (1.653 log 10) or more of the final serum dilution in the serum/virus mixture was regarded as positive.

### 2.9. Statistical Analysis

The data were analyzed using an unpaired *t* test with GraphPad Prism software version 5 (GraphPad Software, San Diego, CA, USA) in which a *p*-value less than 0.05 was considered statistically significant.

## 3. Results

### 3.1. Effect of Chloroform Treatment on FMDV SP and NSP

FMDV type O virus culture supernatant was treated with various concentrations of chloroform, and then Western blot was performed with anti-FMDV type O VP1 and anti-FMDV 3B monoclonal antibodies. Regardless of the chloroform concentration, the intensity of the VP1 (SP) band did not change. However, the 3AB (NSP) band showed different patterns depending on the concentration of chloroform. The 3AB band detected in up to 1% chloroform was not observed in treatment with more than 2% chloroform. However, TCE, an organic solvent structurally similar to chloroform, did not reduce the 3AB band regardless of concentration (Figure 1a). The FMDV titer was measured following chloroform treatment to evaluate the effect of chloroform on viral infectivity. Regardless of chloroform concentration, there was no significant difference in virus titer compared to the untreated virus (Figure 1b).

### 3.2. Comparison of NSPs in FMD Vaccine Antigens Prepared by PEG and Chloroform Treatment

Since chloroform was observed to remove 3AB in vitro, its effect in vivo was investigated. For this purpose, the first and second groups of virus culture supernatants were treated with PEG and chloroform, respectively, to prepare vaccines for animal experiments. The PEG-treated vaccine antigen was resuspended in pellets, resulting in a 10-fold concentration. The chloroform-treated vaccine antigen was concentrated to 15 μg/mL by the ultrafiltration method. The 3AB contained in the vaccine antigens of both groups was analyzed by Western blot. The 3AB band was detected in the PEG group but not in the chloroform group (Figure 2a). To rule out the possibility of 3AB removal during the ultrafiltration process, the FMDV culture supernatant without chloroform treatment was concentrated by the same ultrafiltration. The 3AB band was detected in the retentate fraction rather than in the filtrate after ultrafiltration, and there was no difference in the 3AB band intensity between pre- and post-ultrafiltration (Figure 2b).

### 3.3. Antibody Responses Elicited by Immunizations With PEG- and Chloroform-Treated FMD Vaccines

A total of 10 goats per group were vaccinated three times at 4-week intervals (Figure 3a). All goats were positive for SP antibody with one vaccination, and a PI value of 80% or more was maintained throughout the three vaccinations (Figure 3b). The VN on the homologous virus (O/Boeun/SKR/2017) exhibited similar results to the SP ELISA, with both groups showing a titer of 1:45 (1.653 log 10) and more (Figure 3c). The VN titers were at an average level of 1:375 (2.574 log 10) or more starting from the first vaccination, with no difference between the two groups.

For the NSP antibody, the PEG group was positive for two heads after the second vaccination and positive for four heads after the third vaccination. Meanwhile, none of the goats in the chloroform group showed positivity for the NSP antibody (Figure 3d). In addition, the mean PI value of the chloroform group was significantly lower (*p* = 0.0008) than that of the PEG group after the third vaccination.

## 4. Discussion

FMD vaccine purity relates to the level of FMD NSPs in the final product, which should not induce antibodies that would interfere with serological tests used for serological surveillance of virus circulation in vaccinated populations [5]. PEG is widely used to concentrate and partially purify FMDV to reduce NSPs from FMDV culture supernatants during FMD vaccine production. However, vaccines made with this PEG method inevitably contain some degree of NSPs [7]. In such cases, residual NSPs contained in FMD vaccines could induce NSP antibodies in livestock after repeated vaccination [9]. Accordingly, antibodies against NSP are detected in both FMD vaccinated and naturally infected animals, resulting in invalidation of the DIVA capability. This makes it difficult to continuously monitor the FMDV circulation.

Therefore, we sought to find a new method to selectively remove NSP from FMDV culture supernatant by chemical treatment.

In this study, we used 3AB as a representative protein for the detection of NSPs for the following reasons. First, the most widely used ELISA kit worldwide to detect NSP antibodies is the PrioCHECK FMDV NS kit, and it was designed to detect the 3B epitope [12]. Second, the OIE manual specifies that 3AB and 3ABC are the most reliable indicators of FMDV infection in serological analysis. Third, previous reports showed that 3AB is the major precursor protein containing 3B in FMDV-infected cells [13,14]. As shown in Figure 1, the band pattern of 3AB consisting of 3AB_1_, 3AB_12_, 3AB_123_ by Western blot analysis was the same as a reported previously [13].

The protein 3A is presumed to have a transmembrane domain that could be responsible for the location of the viral RNA replication complex within a membrane context [15]. Therefore, it was postulated that the hydrophobic regions may play a key role in removal by chloroform. However, this scenario was not acceptable because trichloroethylene, structurally similar to chloroform, did not yield the same result. In our preliminary study, other organic solvents such as 1-butanol and 1-pentanol, which are occasionally used to extract antigens from oil adjuvant-based vaccine products, also did not remove 3AB. Further studies are required to elucidate the mechanism of chloroform to remove 3AB from the virus culture supernatant.

In this study, the 146S antigen content of the experimental vaccine was adjusted to 15 μg/mL with reference to a previous study [16]. To adjust the antigen to 15 μg/mL, an ultrafiltration system was used to concentrate the virus culture supernatant. Theoretically, 3AB with about 30 kDa could be easily removed by ultrafiltration with 300 kDa MWCO. However, 3AB was concentrated rather than reduced, as shown in Figure 2b, which was due to the characteristics of tangential flow filtration that is usually employed on an industrial scale. The 3AB protein is also known to exist as a mixture of monomers and dimers due to the cysteine residue in the N-terminal [17]. This may explain the retention of 3AB in the ultrafiltration system. The 3AB protein after chloroform treatment was not detected even after ultrafiltration concentration, indicating its removal by chloroform before ultrafiltration.

Based on the in vitro results, the effect of chloroform treatment was verified by animal experiments. According to the OIE manual 2009, cattle needed to be vaccinated three times and then tested for the presence of NSP antibodies. The vaccine purity was acceptable only if cattle showed negative results in the NSP assays [18]. The OIE guidelines for FMD vaccine purity changed in 2017 so that FMD vaccines were acceptable if less than two of the eight cows were positive for NSP antibodies after being vaccinated twice [5]. In this study, goats were employed rather than cattle for the detection of NSP antibody as a vaccine purity test. This is based on the preliminary data in our institute that goats produced NSP antibodies more rapidly and sensitively than cattle in repeated-vaccination experiments.

While the SP antibody and VN titer were the same between the PEG and chloroform groups, the NSP antibody was detected only in the PEG group. The mean PI value was significantly different between the two groups, indicating that the chloroform removed NSP only without affecting SP or virus particles.

To our knowledge, this is the first report of a new practical method to remove NSPs without eliciting antibodies in livestock after repeated vaccination. This provides several advantages over the conventional FMD vaccine manufacturing process. First, it could yield maximum antigen recovery with little loss of the antigen. When the FMDV is precipitated using the PEG method, the yield of the antigen is reduced by at least 15% [19]. In addition, antigen recovery after the production of the final vaccine on an industrial scale is only about 70% [20]. However, treatment with chloroform had no negative effect on the FMDV particles without risk of antigen loss. Second, it could enable the production of high-quality FMD vaccine antigens at low cost. Other researchers have reported the chromatographic method for FMDV purification and NSP removal [21,22,23], which requires expensive equipment and incurs maintenance costs. However, the chloroform method described in this study requires no additional facility or equipment, and is simple enough for developing countries to make high-purity FMD vaccines by removing NSP at low cost. Third, the production time of the vaccine could be reduced by two days (Figure 4) because the PEG treatment takes at least two days to finish the procedure [22]. Shortened process time by omitting the PEG phase could increase vaccine productivity on an industrial scale.

## 5. Conclusions

The effect of removing NSPs by chloroform treatment was reported for the first time in this study. Chloroform treatment to produce the FMD vaccine antigen was a more efficient method of removing NSPs and yielding the maximum antigen than the PEG method. Therefore, this novel method is expected to revolutionize the conventional process of FMD vaccine production worldwide.

## Figures and Tables

**Figure 1 vaccines-08-00483-f001:**
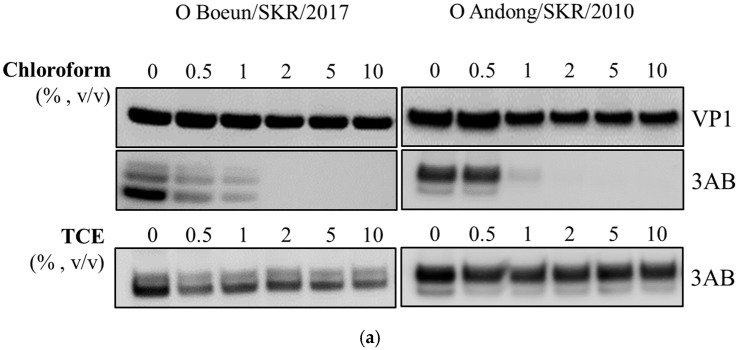
Effect of chloroform treatment on viral proteins of FMD virus (FMDV). (**a**) The O/Boeun/SKR/2017 and O/Andong/SKR/2010 virus culture supernatants were treated with various concentrations of chloroform (upper panel) and trichloroethylene (lower panel). The structural (VP1) and non-structural (3AB) proteins were detected by Western blot analysis using anti-FMDV type O VP1 and 3B monoclonal antibodies. (**b**) The O/Boeun/SKR/2017 (left panel) and O/Andong/SKR/2010 (right panel) virus culture supernatants were treated with various concentrations of chloroform. Viral titers were determined on baby hamster kidney-21 (BHK-21) cells, and the titers were expressed as log10 TCID_50_ per milliliter. All experiments were performed in duplicate.

**Figure 2 vaccines-08-00483-f002:**
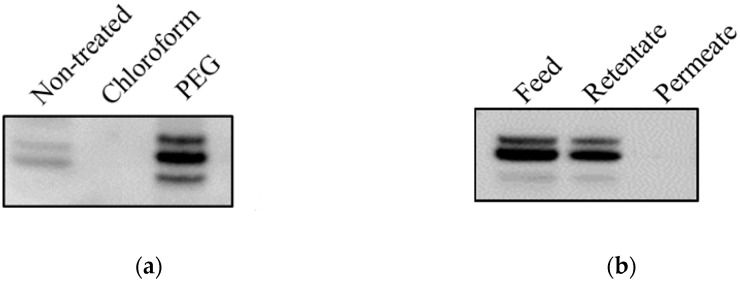
Comparison of non-structural proteins (NSPs) in FMD vaccine antigens prepared by the PEG and chloroform treatment. The O/Boeun/SKR/2017 virus supernatant was treated with 7.5% (*w/v*) of PEG and 10% (*v/v*) of chloroform. The chloroform-treated virus supernatant was subsequently concentrated by ultrafiltration. After the preparation of vaccine antigen, NSP (3AB) was detected by Western blot analysis using anti-FMDV 3B monoclonal antibody. (**a**) FMD vaccine antigens prepared by PEG or chloroform plus ultrafiltration methods. (**b**) The retentate and permeate samples obtained by ultrafiltration of FMDV culture supernatant without chloroform treatment.

**Figure 3 vaccines-08-00483-f003:**
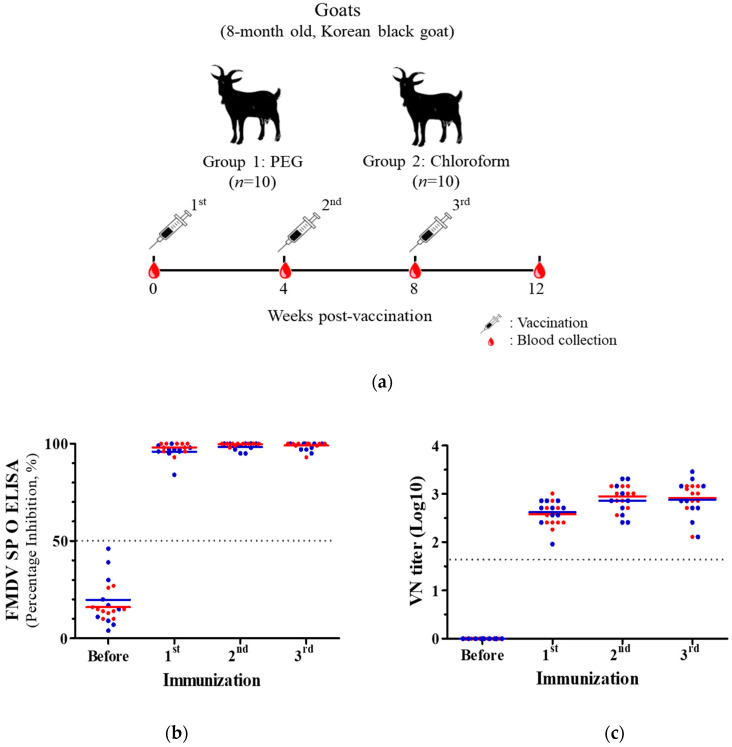
Antibody responses elicited by immunizations with PEG and chloroform treated FMD vaccines. (**a**) Scheme of animal experiment. Goats were divided into 2 groups and vaccinated three times at a 4-week interval with PEG (*n* = 10) and chloroform (*n* = 10) treated FMD vaccines. Blood samples were collected every 4 weeks after vaccination. (**b**) Antibody titers of structural protein were measured by ELISA. The antibody values were expressed as percentage inhibition (PI). PI value of ≥50% (dotted line) was considered positive. (**c**) Homologous virus-neutralizing (VN) antibody titers were measured by virus neutralization test. The VN titers were expressed as a log 10 value. A titer of ≥1.653 log 10 (dotted line) was considered positive. (**d**) Antibody titers of non-structural protein were measured by ELISA. The antibody values were expressed as PI. PI value of ≥50% (dotted line) was considered positive. The full blue and red lines represent the mean of the PI values in each group. A statistically significant difference of the mean values between the two groups in the third vaccination was indicated. *** *p* < 0.001.

**Figure 4 vaccines-08-00483-f004:**
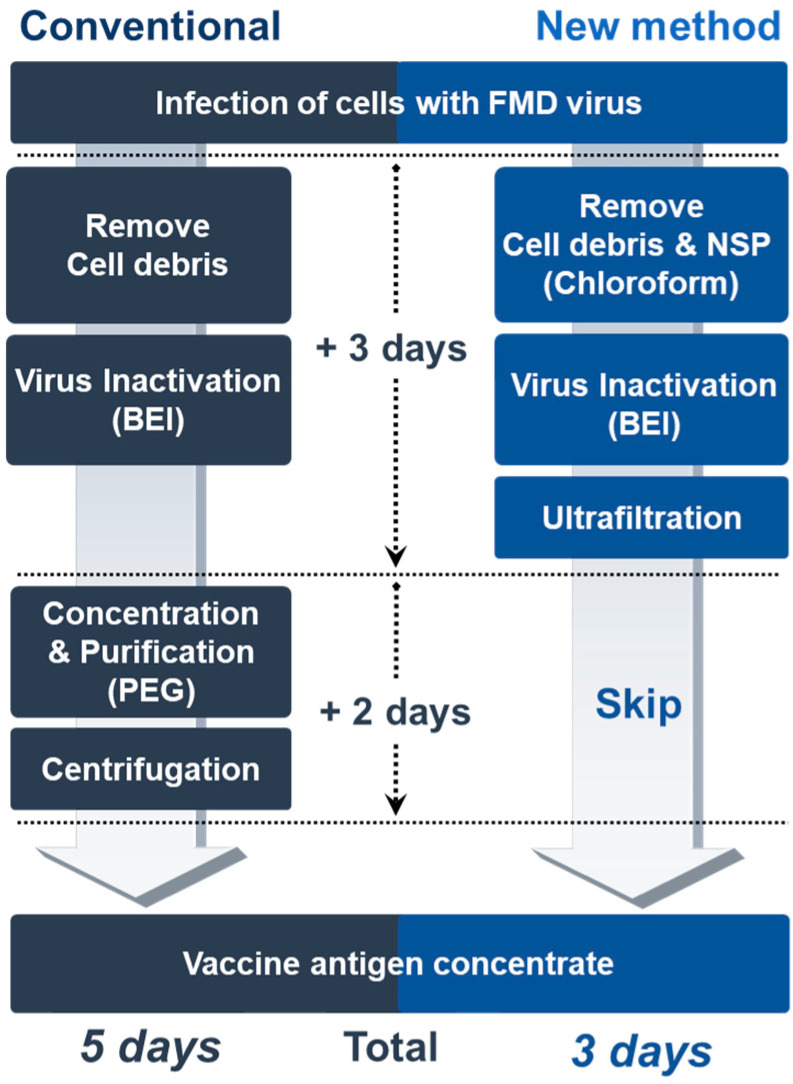
The diagram of FMD vaccine antigen manufacturing process utilizing conventional and new methods. The dotted arrows indicate the turnaround time for the production of FMD vaccine antigen concentrates.

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
