# Peer review of "Efficient Removal of Non-Structural Protein Using Chloroform for Foot-and-Mouth Disease Vaccine Production"

_vaccines, 2020, doi:10.3390/vaccines8030483_

Round 1
Reviewer 1 Report
I found very interesting the results shown in the article "Efficient removal of non-structural protein using chloroform for Foot-and-Mouth Disease Vaccine production". The step for removing the non structural proteins in the vaccines is critical for the production of high quality immunogens that can be use in the control and eradication campaigns. There are no marker vaccines for FMDV, and only inactivated vaccines are allowed to be used. This manuscript offers an enhanced methodology for removing residual non-structural proteins from the viral concentrate, which should help in the differentiation between vaccinated animals and those that have recovered from natural infection.
The experiments carried out to demonstrate the absence of non structural proteins in the concentrate of BHK-21 grown virus, and the invitro tests to show neutralization are clearly displayed, and I think that the experiments are beautifully done. Of course there is a lack of statistical significance of the results, but that is unavoidable in most of the studies requiring animal experiments and I would just mention it as part of the discussion. More importantly, I have some concerns about the conclusions proposed by the authors, which I suggest they should develop more in their discussion, including adding some literature data supporting their claims. No experimental work should be needed, just a deeper literature research. The points of concern are:
1) How much economic and environmental impact may have the general use of ultrafiltration in vaccine production in comparison with the current PEG method. Is it a realistic proposal or not.
2) Given the significant differences in pathogeny and development of the clinical signs, how equivalent is the immunization model with goats in comparison with the cattle model generally used in vaccine studies.
Author Response
Point 1: How much economic and environmental impact may have the general use of ultrafiltration in vaccine production in comparison with the current PEG method. Is it a realistic proposal or not.
Response 1: Ultrafiltration is a commonly used method in the FMD vaccine manufacturing process for the purpose of concentrating viruses. It is easy to perform and takes less time than PEG method that utilize a big equipment such as a continuous centrifuge. But, the ultrafiltration itself does not remove nonstructural proteins as shown in this study. We removed non-structural protein with chloroform and then concentrated viruses up to the required amount of 146S (15 ug/dose). We suggest that this process of using ultrafiltration after chloroform treatment would be a better than PEG method in economic (cost and time) and environmental aspects (requires less space).
Point 2: Given the significant differences in pathogeny and development of the clinical signs, how equivalent is the immunization model with goats in comparison with the cattle model generally used in vaccine studies.
Response 2: It has been reported that antibodies against FMDV are elicited more in goats than in cattle as reported in below reference. In this regard, we chose goat rather than cattle to compare antibody responses against nonstructural protein after repeated vaccination.
* Reference: Lazarus, D.D., et al. Efficacy of a foot-and-mouth disease vaccine against a heterologous SAT1 virus challenge in goats. Vaccine 2020

Reviewer 2 Report
Vaccines-893661
Efficient Removal of Non-structural Protein Using Chloroform for
Foot-and-Mouth Disease Vaccine Production
The work described in this manuscript is important and the experimental design and interpretation of the data are sound. In general, this manuscript is well written but there are a few points that the authors could address. Once these have been attended too, I believe that the manuscript will be suitable for publication
Specifically:
Line 48-50: this sentence is confusing and needs to be re-worded. I would suggest that this long sentence is broken into several shorter statements. simply describe why inactivated vaccine is used, what happens when inactivated vaccines are used and what serological responses are observed with natural infection compared to vaccination with inactivated virus.
Line 52-53: also needs to be changed in line with the previous sentences to explain why elimination of NSPs is important – there is no mention of the DIVA concept – spell it out here – and why it is so important to have a vaccine that provides a DIVA capability. This comment is also relevant to the text at Lines 61-62.
Line 58 – and elsewhere in the text. There are a few small areas where the English expression can be ‘polished’ here – delete “the” from “the PEG” Also Line 119
Line 67 – “in suspension culture” rather than “suspension cells”
Line 73 – was it really a “shaking incubato” or on a shaking platform in an incubator held at 37oC?
Line 74 & 82: that level of precision for the centrifugal force cannot be consistently achieved – approx. 3000g makes more sense
Lines 92,93: dilutions should be expressed as 1/2000 etc 1:2000 means 1 part to 2000 parts (ie 1/2001)
Line 112/113: turn around to “fitted with a p..e… membrane with a MWCO (in full) of 300kDa”
Line 142: what are LFBK cells and their source?
Line 145 & Line 203: it is extremely unusual to express VN titres on a log10 scale – perhaps just state that they were converted to log10 scale for graphical purposes? VN titres are never so precise that they could be determined to 3 decimal places. The practice of expressing titres as a “final” serum dilution (although also used in the OIE manual) is also incorrect – the volume in which the virus is held and also the cells are added does not impact on the quantity of antibodies included in the reaction. In this example, with a 2-fold dilution series, a sample that has a titre of 16 (based on starting dilution of the serum) does not have a titre of 8 when the virus volume is included or 4 when the cells are added. Please do not change as the term used by the authors is consistent with the OIE Manual terminology
Line 222/3 – “head” rather than “heads”
Lines 231-234 – see comments related to Lines 52-53 & DIVA capabilities and mention how the detection of NSP antibodies negates the DIVA capability that an inactivated vaccine may offer.
Finally, while the authors have successfully used chloroform here, they should be aware that this approach may still have problems gaining acceptance in some countries where regulatory authorities are discouraging the use of chloroform or so-treated materials in food producing animals – they want freedom – or when used, convincing evidence that there is no residual chloroform.
Author Response
Response to Reviewer 2 Comments
Point 1: Line 48-50: this sentence is confusing and needs to be re-worded. I would suggest that this long sentence is broken into several shorter statements. simply describe why inactivated vaccine is used, what happens when inactivated vaccines are used and what serological responses are observed with natural infection compared to vaccination with inactivated virus.
Response 1: We have modified them as you requested.
Point 2: Line 52-53: also needs to be changed in line with the previous sentences to explain why elimination of NSPs is important – there is no mention of the DIVA concept – spell it out here – and why it is so important to have a vaccine that provides a DIVA capability. This comment is also relevant to the text at Lines 61-62.
Response 2: We have modified them as you requested. The reason why vaccines with DIVA capability should be developed was added in discussion section.
Point 3: Line 58 – and elsewhere in the text. There are a few small areas where the English expression can be ‘polished’ here – delete “the” from “the PEG” Also Line 119
Response 3: We have corrected as you requested.
Point 4: Line 67 – “in suspension culture” rather than “suspension cells”
Response 4: The BHK 21 suspension cells were transformed from adherent cells. Therefore, we mean suspension cells in contrast to adherent cells. In addition, the words “in suspension culture” has written in line 69 in the same sentence.
Point 5: Line 73 – was it really a “shaking incubato” or on a shaking platform in an incubator held at 37oC?
Response 5: We have corrected as you requested.
Point 6: Line 74 & 82: that level of precision for the centrifugal force cannot be consistently achieved – approx. 3000g makes more sense
Response 6: We have corrected as you requested.
Point 7: Lines 92,93: dilutions should be expressed as 1/2000 etc 1:2000 means 1 part to 2000 parts (ie 1/2001)
Response 7: We have corrected as you requested.
Point 8: Line 112/113: turn around to “fitted with a p..e… membrane with a MWCO (in full) of 300kDa”
Response 8: We have corrected as you requested.
Point 9: Line 142: what are LFBK cells and their source?
Response 9: We explained the details for the cell line.
Point 10: Line 145 & Line 203: it is extremely unusual to express VN titres on a log10 scale – perhaps just state that they were converted to log10 scale for graphical purposes? VN titres are never so precise that they could be determined to 3 decimal places. The practice of expressing titres as a “final” serum dilution (although also used in the OIE manual) is also incorrect – the volume in which the virus is held and also the cells are added does not impact on the quantity of antibodies included in the reaction. In this example, with a 2-fold dilution series, a sample that has a titre of 16 (based on starting dilution of the serum) does not have a titre of 8 when the virus volume is included or 4 when the cells are added. Please do not change as the term used by the authors is consistent with the OIE Manual terminology
Response 10: We have corrected as you requested.
Point 11: Line 222/3 – “head” rather than “heads”
Response 11: We have corrected as you requested.
Point 12: Lines 231-234 – see comments related to Lines 52-53 & DIVA capabilities and mention how the detection of NSP antibodies negates the DIVA capability that an inactivated vaccine may offer.
Response 12: We changed the sentence as your comments.
Point 13: Finally, while the authors have successfully used chloroform here, they should be aware that this approach may still have problems gaining acceptance in some countries where regulatory authorities are discouraging the use of chloroform or so-treated materials in food producing animals – they want freedom – or when used, convincing evidence that there is no residual chloroform.
Response 13: Chloroform is a very small molecule with a molecular weight of 119Da so that it could be sufficiently removed after ultrafiltration with 300kDa MWCO in this study. In addition, the OIE manual says that it is recommended to use an organic solvent for the purpose of removing lipid-enveloped viruses for manufacturing FMD vaccine.

Reviewer 3 Report
In this manuscript Park et al. have developed a method to purify FMD virus for vaccine production from viral nonstructural proteins (NSPs) contained in infected cell supernatants. Removal of viral NSPs from vaccine antigen is necessary so that one can differentiate vaccinated from infected animals. The authors show that chloroform treatment at an appropriate concentration, ie., 2-10%, removes all of the 3AB NSP and does not effect the infectivity of the virus preparation. Furthermore, goats inoculated with up to 3 doses of the chloroform-treated inactivated vaccine do not develop antibodies against NSP 3AB, while some goats inoculated with a PEG treated inactivated vaccine do develop 3AB antibodies. The authors state this procedure is inexpensive and is quicker than the conventional PEG concentration and purification procedure.
Comments:
1. In the Materials and Methods section the authors state that both the PEG treated vaccine and the chloroform treated vaccine were concentrated to 15 ug/ml. Goats were given 2 ml of each vaccine preparation. If my statement is correct each goat was therefore given 30 ug of either vaccine. That is a very large dose. Perhaps 5-10 times more than normally given to cattle/swine. Perhaps the authors should state that this large dose was, presumably, given to test the “cleanliness” of each vaccine and how it would effect the induction of NSP 3AB antibodies . However, in reality this large dose presumably does not reflect the actual amount of virus and contaminating NSPs normally present in administered vaccines even after 2 or 3 inoculations. Please comment.
2. Fig. 4 indicates that virus was first treated with chloroform and then BEI inactivated. However, in the M&M section 2.2 the authors state that inactivated virus was subsequently treated with chloroform. Which is correct?
3. The data from Fig. 1b indicates that chloroform treatment hardly effected the titer of the 2 serotype O viruses examined. However did the authors actually examine the integrity of the virus, by sucrose gradient centrifugation, after BEI and chloroform treatment?
Minor comments:
1. In the following sentence (lines 122-123) - “All goats were vaccinated 3 times every 4 weeks at a dose of 2 mL and bled at 0, 4, 8, and 12 weeks after vaccination to obtain serum samples.” Presumably the authors intended to say that, "....and bled 0, 4, 8, and 12 weeks after the INITIAL vaccination to obtain serum samples."
2. In Section 5 Conclusions line 292-294 I suggest that the authors change the statement that, “….chloroform treatment to produce the FMD vaccine antigen is the most efficient method of removing NSPs and yielding the maximum antigen.” to “….chloroform treatment to produce the FMD vaccine antigen is A MORE EFFICIENT method of removing NSPs and yielding the maximum antigen THEN THE PEG METHOD.”
3. A number of references do not list the journal that the manuscript was published in. For examples references 9, 17, and 20-24.
Author Response
Response to Reviewer 3 Comments
Point 1: In the Materials and Methods section the authors state that both the PEG treated vaccine and the chloroform treated vaccine were concentrated to 15 ug/ml. Goats were given 2 ml of each vaccine preparation. If my statement is correct each goat was therefore given 30 ug of either vaccine. That is a very large dose. Perhaps 5-10 times more than normally given to cattle/swine. Perhaps the authors should state that this large dose was, presumably, given to test the “cleanliness” of each vaccine and how it would effect the induction of NSP 3AB antibodies. However, in reality this large dose presumably does not reflect the actual amount of virus and contaminating NSPs normally present in administered vaccines even after 2 or 3 inoculations. Please comment.
Response 1: FMD vaccine antigens treated with PEG and chloroform were concentrated to 15μg/ml, and experimental vaccines were prepared by adding 1ml of adjuvant. Therefore, the final vaccine immunized with goats contained 15 μg/2ml/dose.
Point 2: Fig. 4 indicates that virus was first treated with chloroform and then BEI inactivated. However, in the M&M section 2.2 the authors state that inactivated virus was subsequently treated with chloroform. Which is correct?
Response 2: The result was the same by both ways. In this study we performed the experiment with inactivated FMDV culture supernatant. The flow diagram shown in Fig.4 is a process to be employed at an industrial scale.
Point 3: The data from Fig. 1b indicates that chloroform treatment hardly effected the titer of the 2 serotype O viruses examined. However did the authors actually examine the integrity of the virus, by sucrose gradient centrifugation, after BEI and chloroform treatment?
Response 3: We observed intact FMDV particle with 25-30 nm in diameter by sucrose density gradient centrifugation after BEI and chloroform treatment. The amount of antigen (146S) was also not affected after BEI inactivation and chloroform treatment.
Minor comments:
Point 4: In the following sentence (lines 122-123) - “All goats were vaccinated 3 times every 4 weeks at a dose of 2 mL and bled at 0, 4, 8, and 12 weeks after vaccination to obtain serum samples.” Presumably the authors intended to say that, "....and bled 0, 4, 8, and 12 weeks after the INITIAL vaccination to obtain serum samples."
Response 4: We have corrected as you requested.
Point 5: In Section 5 Conclusions line 292-294 I suggest that the authors change the statement that, “….chloroform treatment to produce the FMD vaccine antigen is the most efficient method of removing NSPs and yielding the maximum antigen.” to “….chloroform treatment to produce the FMD vaccine antigen is A MORE EFFICIENT method of removing NSPs and yielding the maximum antigen THEN THE PEG METHOD.”
Response 5: We have corrected as you requested.
Point 6: A number of references do not list the journal that the manuscript was published in. For examples references 9, 17, and 20-24.
Response 6: We have corrected as you requested.
